# Prevalence and Contextual Factors of Juvenile Fibromyalgia in a Population-Based Italian Sample of Children and Adolescents

**DOI:** 10.3390/biomedicines11061583

**Published:** 2023-05-30

**Authors:** Susanna Maddali Bongi, Giovanni Vitali Rosati, Guglielmo Bonaccorsi, Chiara Lorini

**Affiliations:** 1Department of Experimental and Clinical Medicine, University of Florence, Largo Brambilla 3, 50134 Florence, Italy; susanna.maddalibongi@unifi.it; 2Local Health Unit Toscana Centro, 50121 Florence, Italy; giovanni.vitalirosati@gmail.com; 3Department of Health Science, University of Florence, Viale GB Morgagni 48, 50134 Florence, Italy; guglielmo.bonaccorsi@unifi.it

**Keywords:** chronic pain, chronic fatigue, population-based, familial aggregation, peer relationship, pediatric headache, difficulties in sleeping, fibromyalgia, children, telehealth

## Abstract

Juvenile Fibromyalgia (JFM) is poorly known, leading to delay in the identification of the syndrome. On the other hand, early diagnosis in children is important to prevent the worsening of the disease. This study aims to estimate the prevalence of JFM in an Italian population-based convenience sample, using different criteria (2010 and 2016 ACR, Yunus and Masi), by involving family pediatricians. It also aims to assess the relationships between JFM and contextual factors of the children and their parents, as well as to raise awareness of JFM among pediatricians. Children’s data were collected using an ad hoc electronic questionnaire. Overall, 7275 questionnaires were collected (48.5% females; mean age: 8.2 ± 3.6 years). Thirty-eight children (0.5%) met the 2010 ACR criteria, and 4 (0.1%) met the 2016 ACR criteria. The likelihood of meeting the 2010 ACR criteria was significantly higher for children older than 8 years (OR: 2.42), those who had injuries during the leisure time that caused persistent pain (OR: 6.49), whose parents (at least one) had a diagnosis of fibromyalgia (OR: 2.54) or diffuse pain (OR: 9.09). In conclusion, 2010 ACR criteria are confirmed as the more appropriate for children and adolescents and the analysis of contextual factors suggests the need for family pediatricians to pay particular attention to the most important predictors of JFM.

## 1. Introduction

Fibromyalgia (FM) is a common disease among adults, particularly among females, that is still underdiagnosed and underestimated [1]. According to recent diagnostic criteria (American College of Rheumatology—ACR) in 2010 and 2016 [2,3], FM is defined as a chronic disease characterized by widespread chronic pain, fatigue, sleep disturbances, somatic and cognitive symptoms, and mood alteration. Since the publication of the 1990 ACR criteria [4], which included only a 3-month history of widespread musculoskeletal pain and the presence of 11 out of 18 tender points, the evolution of the clinical understanding of FM has emphasized the importance of symptoms beyond pain. These symptoms are an essential part of the condition, contribute to overall suffering and distress, and lead to significant disability and a deep impact on a patient’s quality of life. Although the mean age for the FM diagnosis in adults is between 40 and 50 years, many patients report experiencing symptoms during childhood or adolescence [5].

In a clinical study published in 1985, Yunus and Masi described and used the term Juvenile Primary Fibromyalgia Syndrome for the first time [6], and suggested diagnostic criteria based on 33 juveniles aged 17 years or younger, who suffered from chronic pain. Despite that, the clinical complexity of the Juvenile Fibromyalgia (JFM) remains poorly known and understood, leading to delay in the identification and misinterpretation of the syndrome as growing pains, psychosomatic illness, or psychological disorders. The diagnostic process is made difficult by the unavailability of diagnostic tests [7,8]. On the other hand, early diagnosis in children is important to prevent the worsening of the disease and the development of vicious circles that involve—and worsen—pain, mood disorders, and immobility. Finally, it is important to note that JFM symptoms persist over time and the disease stabilizes as adult FM [8,9]. In fact, despite receiving less research attention than adult FM, the clinical characteristics of JFM are substantially similar, indicating that they are the same disease [5].

Children with JFM experience a lower quality of life compared to those with other chronic conditions. This is largely due to the severe psychological distress they presented, including high levels of anxiety, depression, and other emotional disturbances [10]. Additionally, this condition negatively impacts on school performance (with some even experiencing school refusal, or the so-called ‘school rejection syndrome’), and social life (they may become isolated from their classmates). Furthermore, JFM may lead to an early sedentary lifestyle, affecting their physical activity levels [11].

In the few populations studied (Israel, Mexico, Finland, United States, Egypt), the prevalence of JFM among school-aged children is 1.2–6%, based on the 1990 ACR criteria or other questionnaires. JFM typically affects children over the age of four, but it is more common during adolescence (average age of onset: 12–13 years). In these studies, similar to adults, JFM is more prevalent in females (over 80%) and tends to run in families (71% of mothers of children with JFM are also affected by FM) [7,12,13,14]. The reported prevalence rates vary widely, probably due to differences in ethnicity, socio-cultural background, psychological traits of parents, and the study design used [12,13,14,15].

To the best of our knowledge, there is a lack of studies investigating JFM in children and adolescents using population-based samples including both clinical and contextual factors. Moreover, the only Italian study using the 1990 ACR criteria was conducted in 1998 [16]. The study included 2408 children, adolescents, and young adults aged 9 to 21 enrolled at schools, and it found a prevalence of 1.9% among females and 0.6% among males. To date, no studies on JFM have been conducted using the revised ACR diagnostic criteria (2010–2016) in Italy.

To fill in the previously described gaps, this study aims to estimate the prevalence of JFM using the 2010 and the 2016 ACR criteria, as well as Yunus and Masi criteria, in a population-based, convenience sample of children and adolescents, by involving family pediatricians. Moreover, the study aims to assess the relationships between JFM and contextual factors of the children and their parents. By doing so, it aims to make a valuable contribution to the current models concerning this significant illness. Finally, as a secondary aim, our purpose is also to raise awareness of JFM among pediatricians, with the goal of enabling earlier diagnosis and preventing the worsening of symptoms that can lead to the development of FM in adulthood.

## 2. Materials and Methods

### 2.1. Study Design

The study adopted a cross-sectional design and was performed according to the Helsinki Declaration. Data were collected using an ad hoc electronic questionnaire. The aims and the design of the study were shared with a large sample of family pediatricians (*n* = 1500), using the software they usually use to communicate or chat with the parents of their assisted children, as well as to gather information about their patients.

Pediatricians who agreed to participate in the study shared and spread the questionnaire to their patients’ parents through the platform they use. Parents who decided to join the study filled in the questionnaire after providing informed consent. Collected data were extracted by the manager of the platform and shared—in an anonymous format—with the research group. The ad hoc questionnaire was designed to explore by means of the same tool several clinical as well as contextual factors investigated in previous studies. This approach was adopted to obtain a comprehensive understanding of all the potentially impactful factors within a large population-based sample of children.

### 2.2. Questionnaire

The questionnaire was composed of several sections. The first one pertained to the children’s age and gender, as well as data on contextual factors related to the children expressed by a Likert scale to collect the parents’ judgement:relationship with peers (“very good”, “quite good”, “quite bad”, “very bad”, “don’t know”);relationship with parents (“very good”, “quite good”, “quite bad”, “very bad”, “don’t know”);school performance (“very good”, “quite good”, “quite bad”, “very bad”, “don’t know”);being bullied (“yes”, “no”, “don’t know”).

The second one regarded the children’s clinical data, which were used also to identify those who met the criteria to diagnose FM:pain in the previous week, in each of the identified body areas for FM (“yes”, “no”);severity of the pain (VAS: from 0—no pain to 10—very severe);level of severity fatigue, waking unrefreshed, and cognitive symptoms experienced over the previous week using the following scale: “no symptoms”, “slight or mild problems, generally mild or intermittent”, “moderate, considerable problems, often present and/or at a moderate level”, “severe, pervasive, continuous, life-disturbing problems”);other symptoms (fatigue, difficulties in sleeping, difficulties in thinking and remembering, difficulties in studying, headache, anxiety and nervousness, depression or melancholy, diarrhea or constipation, abdominal pain or bloating, itching, rash or hives, skin’s sensitivity to the sun, nausea or heartburn, cold hands and feet, swelling sensation in the hands, palpitations, chest pain, dizziness, shortness of breath, cramps, muscle weakness in the legs, mild fever, painful menstruation, muscle stiffness especially in the morning, shaking of the legs in bed, tingling, numbness, pain modulation by physical activities, pain modulation by weather factors, pain modulation by anxiety or stress; for each, “yes”, “no”);having had road accidents (“yes”, “no”);having had injury in the leisure time that led to persistent pain (“yes”, “no”);having had surgery (“yes”, “no”).

The third section pertained to parents’ contextual and clinical factors:
cohabiting parents (“yes”, “no”);suffering from diffuse pain (“yes”, “no”);having had a diagnosis of fibromyalgia (“yes”, “no”);using or having used psychotropic drugs (“yes”, “no”).

The entire questionnaire can be requested from the corresponding author.

### 2.3. Selection of Children Who Met Fibromyalgia Criteria

Collected data through the questionnaire were used to define the children who tested positive for JFM, and then to calculate the prevalence of JFM. To fulfill the aim of the study, different criteria were considered: 2010 ACR, 2016 ACR, and Yunus and Masi criteria (for the last one, only self-reported information regarding symptoms was considered) [2,3,6].

The Widespread Pain Index (WPI) was calculated by counting the areas in which the patient proved pain over the last week (score between 0 and 19), while the Symptom Severity Scale (SSS) score was obtained by summing the scores of somatic symptoms into a 0–12 scale. In particular, the following somatic symptoms and scores were taken into account for the SSS score:for 2010 and 2016 ACR criteria, waking unrefreshed, cognition, fatigue. For each of them, the level of severity over the past week was considered: “0” for no problem, “1” for slight or mild problems, generally mild or intermittent, “2” for moderate, considerable problems, often present and/or at a moderate level, “3” for severe, pervasive, continuous, life-disturbing problems;for 2010 ACR criteria, somatic symptoms in general: “0” for no symptoms, “1” for few symptoms, “2” for a moderate number of symptoms, “3” for a great deal of symptoms;for 2016 ACR criteria, the number of the following symptoms the patient has been suffering during the previous 6 months (“0” for no, “1” for yes): headaches, pain or cramps in lower abdomen, depression.

For 2010 ACR criteria, a child was considered positive if the following 3 conditions were met:WPI ≥ 7 and SSS score ≥ 5 or WPI ranged between 3 and 6 and SSS score ≥ 9,symptoms have been present at a similar level for at least 3 months,he/she does not have a disorder that would otherwise explain the pain.

On the contrary, a child satisfies 2016 ACR criteria if the following 3 conditions were met:WPI ≥ 7 and SSS score ≥ 5 or WPI ranged 4–6 and SSS score ≥ 9,generalized pain, defined as pain in at least 4 of 5 regions (not including jaw, chest, and abdominal pain),symptoms have been lasting for at least 3 months.

Finally, due to the study design, tender points examination was not performed, so only self-reported symptoms (minor criteria) were considered with respect to Yunus and Masi criteria. These symptoms included: chronic anxiety or tension; fatigue; poor sleep; chronic headache; irritable bowel syndrome; subjective soft tissue swelling; numbness; pain modulation by physical activity; pain modulation by weather factors; pain modulation by anxiety/stress. To meet the Yunus and Masi criteria, a minimum of 3 symptoms has to be reported.

### 2.4. Statistical Analyses

Data were presented as percentages or as mean (±standard deviation) and median (interquartile range—IQR).

Cohen’s kappa was used to assess the agreement between the classification resulted applying the two criteria for FM (2010 ACR and 2016 ACR). Kappa values of 0.01–0.20, 0.21–0.40, 0.41–0.60, 0.61–0.80, 0.81–0.99, and 1 represented slight, fair, moderate, substantial, almost perfect, and perfect agreement, respectively.

Bivariate analysis was conducted to evaluate associations between 2010 ACR or 2016 ACR criteria and the following contextual factors: sex, age, relationship with peers, school performance, being bullied, relationship with parents, cohabiting parents, at least one parent with diffuse pain, at least one parent with fibromyalgia, at least one parent who is using or used psychotropic drugs, having had road accidents, injuries during the leisure time that caused persistent pain, or surgeries. Associations were tested using Fisher exact test. Similar analysis was not performed considering positivity to Yunus and Masi criteria because of the limit in applying those criteria in our study (due to the study design, only minor criteria were detected).

Then, a multivariate logistic regression analysis was performed considering positivity to 2010 ACR criteria as outcome (dependent) variable (“yes” vs. “no”) and all the contextual variables significantly associated—at the bivariate analysis—with 2010 ACR criteria as predictors (independent variables). The backward stepwise method was applied to obtain the final model. Similar analysis was not performed considering positivity to 2016 ACR criteria due to the low number of children who met those criteria.

All the analyses were conducted using IBM SPSS version 28.0.1.0 (IBM Corp., New York, NY, USA; formerly SPSS Inc., Chicago, IL, USA), considering 0.05 as the alfa level.

## 3. Results

### 3.1. Prevalence of JFM

Overall, 47 pediatricians from 11 out of 20 Italian regions joined the study, and 7275 questionnaires were collected. Children were equally distributed by sex (51.5% males, 48.5% females), and had a mean age of 8.2 ± 3.6 years (median: 8 years, IQR: 5–11).

Out of all children, 38 (0.5%) met the 2010 ACR criteria, and 4 (0.1%) met the 2016 ACR criteria. Additionally, all the children who met the 2016 ACR criteria also met the 2010 ACR criteria. The agreement between the two criteria was poor (Cohen’s kappa = 0.081; “slight” agreement). Furthermore, 46 children who met the Yunus and Masi criteria were investigated in the questionnaire (only self-reported symptoms, minor criteria).

Table 1 presents the demographic and clinical characteristics of the entire sample, as well as by positivity to ACR criteria. Sex was not associated with JFM, regardless of whether the 2010 or 2016 ACR criteria were used. On the other hand, those who met the 2010 ACR criteria were significantly older than those who tested negative: their mean age was 11.9 ± 3.6 years (IQR: 9–14.2) and 81.6% were older than 8 years.

For the majority of children (86.3%), no pain over the previous week was reported. Children who tested positive for ACR criteria reported more frequent pain in almost all body areas, compared to children who tested negative. Specifically, when considering 2010 ACR criteria, children with JFM reported more frequent pain in all their body areas, except for the right lower arm. Conversely, when considering the 2016 ACR criteria, no significant differences were observed for both arms, as well as for the jaw. Legs and abdomen were the areas in which pain was reported more frequently among children positive for 2010 ACR criteria, whereas legs, abdomen, and upper arms among children positive for 2016 ACR criteria. Among children who met the 2010 ACR criteria, 13 (34.2%) reported pain in 3 body regions, 12 (3.4%) in 4 body regions, 4 (10.5%) in 5 body regions, and 5 in 6 or more body regions. Among children who met the 2016 ACR criteria, 3 (75%) reported pain in 7 or more body regions.

Out of the entire sample, 711 children (9.8%) reported fatigue, 1551 (21.3%) experienced waking up unrefreshed, and 1074 (14.8%) reported cognitive symptoms (Table 1). Waking up unrefreshed was reported to be the most severe symptom, with 3.9% of children experiencing moderate or severe problems the week before. In contrast, fatigue was reported to be the least severe symptom, with only 0.1% experiencing severe symptoms and 1.5% moderate problems. When considering children who met 2010 ACR criteria, fatigue was reported in 37 cases (97%), waking unrefreshed in 36 (94.7%), and cognitive symptoms in 34 (89.5%), while considering 2016 ACR criteria, all the children reported the three symptoms. Waking unrefreshed was the symptom with the highest level of severity (Table 1).

Over the past six months, among children who met 2010 ACR criteria, 89.5% reported experiencing headaches, 86.8% laziness or depressive symptoms, and 26.2% abdominal cramps. All children who met 2016 ACR criteria reported experiencing headaches, while three out of four reported laziness or depressive symptoms and abdominal cramps.

Table 2 shows the descriptive statistics of WPI, SSS score, VAS score, and symptom count for the whole sample and by subgroups based on ACR criteria. As expected, children with JFM presented significantly higher values compared to those without the disease. In particular, among children who met the 2010 ACR criteria, the mean WPI and SSS scores were 4.5 (SD: 1.6) and 7.5 (SD: 1.4), respectively, while for those who met the 2016 ACR criteria, they were 6.5 (SD: 1.7) and 7.2 (SD: 1.3), respectively.

Regarding pain severity measured by the VAS score, 30 children who met the 2010 ACR criteria (79%) and all the children who met the 2016 ACR criteria reported a score of 5 or higher.

As far as somatic symptoms were concerned, significant differences were observed between children with or without JMF according to 2010 ACR criteria, with all investigated symptoms being reported more frequently among those who met the criteria, except for rash/hives and skin’s sensitivity to the sun (Table 3). However, considering 2016 ACR criteria, statistically significant associations were found only for 9 out 27 somatic symptoms (Table 3). Among children with JFM, the mean of symptom count was higher than 11 (12.5 ± 4.5 considering 2010 ACR criteria; 11.2 ± 1.7 considering 2016 ACR criteria), while for those who did not meet the criteria, it was about 2.6 (Table 2).

Fatigue, headache, anxiety, and nervousness were the most frequent (>85%) somatic symptoms among children with JFM (Table 3).

### 3.2. Association with Contextual Factors: Predictors of Compliance with ACR Criteria

The association between the classification of the children according to the criteria for FM (2010 ACR, 2016 ACR) and contextual factors is described in Table 4. The classification based on the 2010 ACR criteria was significantly associated with most of the investigated variables. In particular, children who met the FM criteria had worse relationships with peers or with parents, worse school performance, a higher frequency of being bullied, involvement in leisure time accidents (with persistent pain), and a history of surgeries. Additionally, their parents had a higher frequency of not cohabiting, presented diffuse pain, had a diagnosis of fibromyalgia, and/or used or had used psychotropic drugs.

In Table 5, the results of multivariate logistic regression analysis are reported (final model, obtained using the backward stepwise procedure). As a consequence of bivariate analyses, one model was performed, considering agreement with 2010 ACR criteria as the outcome variable, and age, relationship with peers, school performance, being bullied, cohabiting parents, relationship with parents, at least one parent with diffuse pain, at least one parent with fibromyalgia, at least one parent who uses or has used psychotropic drugs, having had injuries on the leisure time that caused persistent pain, or surgeries. In the final model, the likelihood of meeting the 2010 ACR criteria was significantly higher for children older than 8 years (OR: 2.42), those who had injuries during the leisure time that caused persistent pain (OR: 6.49), whose parents (at least one) had a diagnosis of fibromyalgia (OR: 2.54), or diffuse pain (OR: 9.09).

## 4. Discussion

To the best of our knowledge, this is the first study investigating the JFM prevalence in a large population-based sample of children and adolescents using the recent ACR criteria, considering both versions of 2010 and 2016, as well as self-reported symptoms (minor criteria) of Yunus and Masi criteria.

Although the 1990 ACR criteria were used to evaluate JFM prevalence in some target groups, they were never validated in the pediatric population. The 2010 ACR criteria, designed for the diagnosis of FM in adults, were evaluated to be used for the diagnosis of JFM in adolescent females, with the Yunus and Masi criteria as the gold standard, with a sensitivity of 89.4% and a specificity of 87.5%, suggesting that they can be applied to this population. Conversely, the 2016 ACR adult criteria have not been studied yet for use in the pediatric FM population.

Compared to the Yunus and Masi criteria, the 2010 ACR criteria have many advantages; in fact, they guarantee an easy and fast assessment of symptoms, and the exclusion of the tender points examination, which are not always present in the disease and are difficult to evaluate. Moreover, the absence of the tender points examination fosters the implementation of population studies on large samples such as ours, based on a self-compiled questionnaire to collect data instead of clinical evaluations carried out during medical examinations, which inevitably leads to a smaller numerousness.

The design of this study allows the comparison between different criteria for FM. Indeed, our results suggest that the 2010 preliminary criteria are more suitable for use in the pediatric population than the 2016 criteria, indicating that the assessment of the extent of somatic symptoms is helpful in this target group. In fact, 38 (0.5%) children met the 2010 ACR criteria, and only 4 (0.1%) the 2016 ACR criteria. Moreover, all the children who fulfilled the 2016 ACR criteria, also met the 2010 ACR criteria. Regarding Yunus and Masi criteria, for 46 (0.6%) children, 3 self-reported symptoms (minor criteria) have been referred; however, the diagnosis of JFM must be confirmed by physical examination looking for the presence of at least 5 tender points.

According to 2010 ACR criteria, in our study, the prevalence of JFM is 0.5%, lower than those reported by other authors using the 1990 ACR criteria [12,13,16,17], and in a retrospective study using the International Classification of Disease, ninth version (ICD IX) [14]. This variability may be due to racial and sociocultural differences between populations, as well as to the study design applied. Moreover, in comparing prevalence described in different studies, the age range of the investigated sample has to be taken into account. In fact, in our study, the children/adolescents ranged between 1 and 17 years old, with a large group of children under the age of 8 (81.6%), which is different from previous studies which included only 9 to 15 years old population. As a matter of fact, JFM is rare among children younger than 9, and the average age of diagnosis in children and adolescents is around 13–15 years [3,6,18,19]. The statistically significant association we have found between age and positivity for the ACR criteria—either 2010 or 2016—confirms the above concerns. What seems to emerge is that the 2010 ACR criteria are more selective for the diagnosis of JFM than those of 1990.

Differently from other studies conducted in adult or pediatric populations [12,13,16,20], the prevalence of JFM in our survey does not statistically differ between sexes, although it tends to be higher among females (according to the 2010 ACR criteria: 0.6% among females; 0.4% among males). These data agree with the progressive reduction of the female/male ratio in the incidence of FM in studies that do not use tender points as a criterion [21].

Almost all the symptoms investigated by our questionnaire (somatic symptoms of the ACR 2010 criteria and the Yunus and Masi questionnaire) were reported by our patients. Table 6 describes the percentage of symptoms detected among the children of our sample who met the 2010 ACR criteria in comparison with those described in a study by Wolfe (1990) conducted in adults, as well as in other surveys performed in children with JFM.

The rate of widespread pain is similar in all studies (93–100%). On the contrary, fatigue is more frequent in our work (94.7%) with respect to that reported in the other studies; in particular, it was much more frequent than that described by Siegel (62%), Eraso, and Gedalia (20–28%) [20,23], and slightly higher than those reported by Yunus (91%) and Wolfe (81%). Sleep disorders (68.4%), headache (78.9%), and abdominal symptoms (26.2%) tend to be as frequent as reported in the other studies. The exceptions were the percentage of sleep disturbances, which was higher in Siegel’s work (96%) and headache, which was lower in the studies by Wolfe and Yunus (53 and 58%, respectively). Instead, depression (57.9%) and anxiety (86%) are significantly more represented in our study than in the other surveys, especially with respect to the Eraso and Gedalia studies.

Our data confirm that fatigue and depression are very common symptoms among patients with FM, supporting the particular prominence given in the 2010 and 2016 ACR diagnostic criteria [3], together with sleep disorders, headache, and abdominal symptoms. In addition, cognitive symptoms, which are very frequent in our casuistry (difficulties in thinking and remembering: 36.8%; difficulties in studying: 63.2%), are relevant in the recent criteria [3] while they were not investigated in the other studies that used different criteria for JFM.

Subjective sensations such as soft tissue swelling, stiffness, and numbness are reported with a wide variability in frequency (Table 6), confirming the difficulty in detecting them in the pediatric population.

The variability in the prevalence of somatic symptoms reported by other authors, as well as the similarities and the differences with respect to our data (Table 6) can be due to the diversity of the population studied (primarily, the age range), the items included into the questionnaires used as survey tools, and the methods for data collection (for example, online questionnaire, telephone interview, face-to-face interview conducted in the clinic). Nonetheless, the studies confirm the multiplicity of symptoms in JFM and the complexity of its diagnosis.

As far as the multiplicity of symptoms is concerned, in our study, several symptoms appear to be very common, as follows (in order of frequency): dysmenorrhea (80% among females over 10 years), diarrhea/constipation (78.8%), muscle weakness in the legs (60.5%), cold hands and feet (44.7%), itching (42.1%), chest pain (39.5%), nausea/heartburn (39.5%), cramps (39.5%), mild fever (36.8%), shaking of the legs in bed (36.8%), shortness of breath (28.9%), tingling (28.9%), rash/hives (26.3%), palpitations (21.1%), and dizziness (21.1%). Moreover, all these symptoms are significantly more frequent among children who met the 2010 ACR criteria, with the exclusion of rash/hives.

Therefore, the data from our study do not allow us to be totally in agreement with the recommendation of Ting [17] concerning the removal of some items from the list of somatic symptoms such as chest pain, fever, cold hands and feet, and vomiting. On the other hand, we agree on the deletion of other items such as oral ulcers, loss of/change in taste, seizures, hearing difficulties, hair loss, painful urination, hives/welts, sun sensitivity, and bladder spasms.

Pain is reported by 100% of children who met 2010 ACR criteria; considering the severity, the mean WPI score was 4.5 (±1.5), a little lower than that reported by Ting [17] in 97 children with JFM using the 2010 ACR criteria (6.3 ± 1.6), and by Wolfe [2] in a sample of adults with FM (6.5 ± 2.3). Abdomen (52.6%) and legs (47% left and 52.6% right) are the area where pain was most frequently reported. These areas correspond perfectly to the most frequent painful sites reported by Yunus [6] who considers the joints of the limbs (knees and ankles) and does not take into consideration the abdomen. In this regard, we observe that among children for whom the diagnosis of JFM has been delayed, the pain in the lower limbs is often interpreted as “growing pains” [8,24].

According to our data, neck (31.6%), upper and lower back, upper arm, and chest (28.9%) are the body areas in which pain was reported for a greater number of children who met the 2010 ACR criteria, corresponding (in order of frequency) to elbows, lower and upper back, wrists, cervical spine, trapezius, and hands as reported in the Yunus study.

To our knowledge, only the study conducted by Ting has been performed among children using 2010 ACR criteria [17]; in that study, 47 girls aged 11–17 years (mean age 15.3) recruited by the Rheumatology Clinic of Cincinnati were included. SSS score calculated in our study is higher than that described by Ting (7.5 vs. 5.75), while WPI is lower (4.5 vs. 10.9) and symptom count is similar (12.5 vs. 14.4); in particular, in our sample, more children have mild and moderate levels of symptoms and fewer have severe ones. These differences may be due to the younger age of our group, as well as of the study design: in our study, a population-based sample has been included while in that conducted by Ting, the children have been enrolled in a Rheumatology Clinic.

Therefore, our results obtained using different criteria for FM (Yunus and Masi, 2010 and 2016 ACR criteria for FM) allow us to conclude, in agreement with Ting, that the 2010 ACR criteria offer a simplified means to diagnose a condition as complex as JFM. Compared to the previous classification criteria, these criteria have several advantages as they are based only on the 2010 questionnaire, which can be quickly filled in also by the parents and therefore easily used either in a clinical setting, or in an epidemiological population-based study. Furthermore, the 2010 criteria, unlike the 1990 ACR criteria, include the assessment of a large number of important symptoms for the JFM diagnosis, as well as the severity of the main ones: fatigue, unrefreshing sleep, and cognitive symptoms. In the end, compared to the later 2016 criteria, our data indicate that the questionnaire based on 2010 criteria, that includes multiple symptoms, is more comprehensive and gives more complete and useful information to diagnose JFM.

In this study, contextual factors of the children (relationship with peers, relationship with parents, school performance, and being bullied), previous painful traumatic situations (road accidents, injury in the leisure time, surgery), and contextual and clinical factors of the parents (cohabiting parents, to have diffuse pain, having had a diagnosis of fibromyalgia, to use or have used psychotropic drugs) have also been investigated. Bivariate analysis revealed that JFM diagnosis according to 2010 ACR criteria is significantly associated with most of the investigated variables. In particular, children who met the criteria for FM had significantly worse relationships with peers or with parents, had worse school performance, have been more frequently bullied, involved in leisure time accidents (with persistent pain), and had surgeries. Moreover, their parents more frequently are not cohabitants, presented diffuse pain, had a diagnosis of fibromyalgia, or use or had used psychotropic drugs. The results confirm what have been reported by other authors.

Familial aggregation is a well-documented feature of FM [25,26]. Roizenblatt et al. (1997) [27] studied 34 children with JFM aged 11 (±1) years and their mothers, observing a significant predominance of mothers with FM in the group of children with JFM (71%) compared to children with diffuse pain (30%) and asymptomatic children (0%). In addition, a significant correlation between polysomnographic indexes, sleep anomalies, and pain manifestations was present in children and their mothers.

Nelson et al. (2017) [28] studied 86 young adults with JFM and found that 37% of them reported a trauma history. JFM participants with and without a trauma history did not significantly differ for pain and physical functioning, but JFM participants with a trauma history were significantly more likely to have psychological comorbidities.

Hypermobility syndrome is a well-known clinical association of JFM [29]. The most probable hypothesis is that hyperlaxity leads to recurrent micro-trauma and occasional joint dislocation. The resulting recurrent peripheral pain may lead to the development of central sensitization and then JFM. Consistently with our results, in the study of Imbierowicz and Egle (2001) [30], the patients with FM, compared to patients with other pain disorders, showed the highest score of childhood adversities. In addition to sexual abuse and physical maltreatment, the FM patients reported more frequently a poor emotional relationship with both parents, a lack of physical affection, experiences of parents’ physical quarrels, as well as alcohol or other problems of addiction of the mother, separation, and a poor financial situation before the age of seven.

Moreover, in the study by Schanberg et al. [31], parents of children with JFM reported multiple chronic pain conditions, including FM and their pain history, and the family environment correlated with the health status of adolescents with JFM.

Kashikar-Zuch et al. (2008) [32] detected JFM family relationships were characterized by higher levels of conflict, lower levels of cohesion, and less organizational structure than comparison families. Furthermore, mothers of adolescents with JFM reported significantly greater depressive symptoms than mothers of comparison peers.

In another work, Kashikar-Zuch et al. (2007) [33] assessed the peer relationships of adolescents with JFM and demonstrated that, from the perspective of peers, they are consistently rated as more sensitive/isolated than their healthy peers, having significantly fewer friends, fewer reciprocated friendships, and being less well liked.

Multivariate analysis confirms the significant associations between some of the contextual factors investigated. In particular, the likelihood of meeting the 2010 ACR criteria was significantly higher for children older than 8 years old (OR: 2.42), among those who had injuries during the leisure time that caused persistent pain (OR: 6.49), whose parents (at least one) had a diagnosis of fibromyalgia (OR: 2.54), or had diffuse pain (OR: 9.09). On the contrary, relationships with peers and with parents, school performance, being bullied, cohabiting parents, and the use of psychotropic drugs by parents were excluded during the stepwise procedure, probably due to collinearity with other variables.

This study has some limitations. First, a convenience sample has been included, since the questionnaire was sent only to the parents of the children assisted by family pediatricians who joined the study. Moreover, the participation of the parents was voluntary. Therefore, we cannot exclude a selection bias for both—family pediatricians and parents—that may have affected the results. Additionally, data were collected through an online self-reported questionnaire filled in by the parents to collect data regarding the health status of the children as well as their conditions. In this perspective, recall bias, lack of knowledge regarding mild symptoms of the children, and social desirability bias could have affected some of the results. On the other hand, the study design, in particular the use of an online self-reported questionnaire, allowed us to recruit a large sample size.

## 5. Conclusions

Our study made it possible to calculate the prevalence of JFM in a large population-based sample of children and adolescents, by means of family pediatricians who used a telehealth medicine system based on a digital platform. Despite the many limitations, the results provide some useful information for the diagnosis and the prevention of JFM. First, 2010 ACR criteria are confirmed as the more appropriate for children and adolescents, and the ease and quickness of use even by non-specialists move toward their applicability in pediatrician clinical practice, at least as an initial screening tool for JFM. Second, the analysis of contextual factors in the results suggests the need for family pediatricians to pay particular attention to the most important predictors of JFM, in order to prevent the syndrome or to anticipate the diagnosis.

In particular, we strongly advise family pediatricians not to underestimate children experiencing recurring pain that cannot be attributed to an organic cause. It is crucial to investigate whether these children exhibit any of the common symptoms associated with fibromyalgia, such as fatigue, sleep difficulties, headaches, or abdominal discomfort. We also recommend pediatricians to inquire about the following factors: non-cohabitant parents, presence of diffuse pain, previous diagnosis of fibromyalgia, past or current use of psychotropic drugs, strained relationships with peers or parents, poor academic performance, history of bullying, involvement in leisure time accidents, and past surgeries.

This study also emphasizes the potential of implementing telehealth and digital platforms in providing proactive healthcare. These platforms enable patients to be questioned about their overall health, unveiling potential underdiagnosed conditions.

Finally, we hope further works on children suffering from chronic pain syndromes will provide pediatricians with tools to better diagnose and effectively treat this underserved population, so as to prevent the unrecognition of JFM and to avoid the fast progression towards the more severe forms of FM in adulthood.

## Figures and Tables

**Table 1 biomedicines-11-01583-t001:** Demographic and clinical data of the sample.

Variables		Whole Sample *n* = 7275 *n* (%)	2010 ACR			2016 ACR		
			Positive *n* = 38 *n* (%)	Negative *n* = 7237 *n* (%)	*p* *	Positive *n* = 4 *n* (%)	Negative *n* = 7271 *n* (%)	*p* *
Sex	Male	3750 (51.5)	16 (42.1%)	3734 (51.6)	0.258	1 (25%)	3749 (51.6)	0.360
	Female	3525 (48.5)	22 (57.9%)	3503 (48.4)		3 (75%)	3522 (48.4)	
Age	<8 years old	3521 (48.3)	7 (18.4%)	3505 (48.4)	<0.001	3511 (48.3)	1 (25%)	0.626
	≥8 years old	3763 (51.7)	31 (81.6%)	3732 (51.6)		3760 (51.7)	3 (75%)	
Areas with pain over the previous week	Shoulder girdle, left	32 (0.4)	5 (13.2)	27 (0.4)	<0.001	2 (50)	30 (0.4)	<0.001
	Shoulder girdle, right	38 (0.5)	5 (13.2)	33 (0.5)	<0.001	2 (50)	36 (0.5)	<0.001
	Upper arm, left	36 (0.5)	5 (13.2)	31 (0.4)	<0.001	3 (75)	33 (0.5)	<0.001
	Upper arm, right	33 (0.5)	11 (28.9)	22 (0.3)	<0.001	3 (75)	30 (0.4)	<0.001
	Lower arm, left	6 (0.1)	1 (2.6)	5 (0,1)	0.031	0 (0)	6 (0.1)	1
	Lower arm, right	11 (0.2)	1 (2.6)	10 (0.1)	0.056	0 (0)	11 (0.2)	1
	Hip (buttock, trochanter), left	23 (0.3)	4 (10.5)	19 (0.3)	<0.001	1 (25)	22 (0.3)	0.013
	Hip (buttock, trochanter), right	32 (0.4)	6 (15.8)	26 (0.4)	<0.001	1 (25)	31 (0.4)	0.017
	Upper leg, left	48 (0.7)	8 (21.1)	40 (0.6)	<0.001	1 (25)	47 (0.6)	0.026
	Upper leg, right	54 (0.8)	9 (23.7)	45 (0.6)	<0.001	1 (25)	53 (0.7)	0.029
	Lower leg, left	253 (3.5)	18 (47.4)	235 (3.2)	<0.001	3 (75)	250 (3.4)	<0.001
	Lower leg, right	291 (4)	20 (52.6)	271 (3.7)	<0.001	3 (75)	288 (4)	<0.001
	Jaw, left	22 (0.3)	6 (15.8)	16 (0.2)	<0.001	1 (25)	21 (0.3)	0.012
	Jaw, right	18 (0.2)	5 (13.2)	13 (0.2)	<0.001	0 (0)	18 (0.2)	1
	Chest	112 (1.5)	11 (28.9)	101 (1.4)	<0.001	0 (0)	112 (1.5)	1
	Abdomen	264 (3.6)	20 (52.6)	244 (3.4)	<0.001	3 (75)	261 (3.6)	<0.001
	Upper back	57 (0.8)	11 (28.9)	46 (0.6)	<0.001	2 (50)	55 (0.8)	<0.001
	Lower back	98 (1.3)	11 (28.9)	87 (1.2)	<0.001	2 (50)	98 (1.3)	1
	Neck	124 (1.7)	12 (31.6)	112 (1.5)	<0.001	0 (0)	124 (1.7)	1
Fatigue	No symptoms	6564 (90.2)	1 (2.6)	5916 (81.7)	<0.001	0 (0)	5917 (81.4)	0.001
	Slight or mild problems, generally mild or intermittent	593 (8.2)	16 (42.1)	578 (8)	<0.001	2 (50)	592 (8.1)	0.036
	Moderate, considerable problems, often present and/or at a moderate level	107 (1.5)	20 (52.6)	87 (1.2)	<0.001	2 (50)	105 (1.4)	0.001
	Severe, pervasive, continuous, life-disturbing problems	10 (0.1)	1 (2.6)	9 (0.1)	0.051	0 (0)	10 (0.1)	1
Waking unrefreshed	No symptoms	5724 (78.7)	2 (5.3)	5053 (69.8)	<0.001	0 (0)	5054 (69.5)	0.088
	Slight or mild problems, generally mild or intermittent	1265 (17.4)	8 (21.1)	1261 (17.4)	0.668	2 (50)	1269 (17.5)	0.607
	Moderate, considerable problems, often present and/or at a moderate level	241 (3.3)	22 (57.9)	219 (3)	<0.001	1 (25)	239 (3.3)	0.006
	Severe, pervasive, continuous, life-disturbing problems	45 (0.6)	6 (15.8)	39 (0.5)	<0.001	1 (25)	44 (0.6)	0.025
Cognitive symptoms	No symptoms	6201 (85.3)	4 (10.5)	5490 (75.9)	<0.001	0 (0)	5494 (75.6)	0.004
	Slight or mild problems, generally mild or intermittent	831 (11.4)	17 (44.7)	815 (11.3)	<0.001	2 (50)	830 (11.4)	0.067
	Moderate, considerable problems, often present and/or at a moderate level	206 (2.8)	12 (31.6)	195 (2.7)	<0.001	2 (50)	205 (2.8)	0.005
	Severe, pervasive, continuous, life-disturbing problems	37 (0.5)	5 (13.2)	32 (0.4)	<0.001	0	37 (0.5)	1

* Fisher exact test.

**Table 2 biomedicines-11-01583-t002:** Descriptive statistics of WPI score, SS score, pain (VAS score), and symptom count.

Variables	2010 ACR Criteria					2016 ACR Criteria					Whole Sample	
	Positive *n* = 38		Negative *n* = 7237		*p* *	Positive *n* = 4		Negative *n* = 7271		*p* *		
	Mean ± SD	Median (IQR)	Mean ± SD	Median (IQR)		Mean ± SD	Median (IQR)	Mean ± SD	Median (IQR)		Mean ± SD	Median (IQR)
WPI	4.5 ± 1.6	4 (3–5)	0.2 ± 0.6	0 (0)	<0.001	6.5 ± 1.7	7 (4.7–7.7)	0.2 ± 0.6	0 (0)	<0.001	0.2 ± 0.65	0 (0)
SSS score	7.5 ± 1.4	7 (7–8)	1.2 ± 1.5	1 (0–2)	<0.001	7.2 ± 1.3	7 (6.2–8.5)	1.2 ± 1.6	1 (0–2)	<0.001	1.2 ± 1.6	1 (0–2)
VAS score	5.8 ± 1.4	6 (5–7)	3.3 ± 1.8	3 (2–5)	<0.001	6.2 ± 0.9	6.5 (5.2–7)	3.3 ± 1.8	3 (2–5)	0.003	3.3 ± 1.78	3 (2–5)
Symptom count	12.5 ± 4.5	11 (9.7–15)	2.6 ± 2.5	2 (1–4)	<0.001	11.2 ± 1.7	11.5 (9.5–12.7)	2.7 ± 2.6	2 (1–4)	<0.001	2.7 ± 2.6	2 (1–4)

SD—standard deviation; IQR—interquartile range; WPI—widespread pain index; SS—symptom severity scale; VAS—visual analogue scale. * Mann–Whitney test.

**Table 3 biomedicines-11-01583-t003:** Somatic symptoms reported in the last 3 months.

Somatic Symptoms	Whole Sample *n* = 7275 *n* (%)	2010 ACR			2016 ACR		
		Positive *n* = 38 *n* (%)	Negative *n* = 7237 *n* (%)	*p* *	Positive *n* = 4 *n* (%)	Negative *n* = 7271 *n* (%)	*p* *
Fatigue	2739 (37.6)	36 (94.7)	2192 (30.3)	<0.001	3 (75)	2224 (30.6)	0.009
Difficulties in sleeping	1513 (20.8)	26 (68.4)	1487 (20.5)	<0.001	1 (25)	1512 (20.8)	1
Difficulties in thinking and remembering	398 (5.5)	14 (36.8)	384 (5.3)	<0.001	1 (25)	397 (5.5)	0.202
Difficulties in studying	689 (9.5)	24 (63.2)	665 (9.2)	<0.001	3 (75)	686 (9.4)	0.003
Headache	2213 (28.5)	34 (89.5)	2038 (28.2)	<0.001	4 (100)	2209 (30.4)	0.009
Anxiety and nervousness	1529 (21)	33 (86.8)	1495 (20.7)	<0.001	3 (75)	1525 (21)	0.031
Depression, melancholy	214 (2.9)	22 (57.9)	192 (2.7)	<0.001	2 (50)	212 (2.9)	0.005
Diarrhea, constipation	3004 (41.3)	30 (78.9)	2974 (41.1)	<0.001	3 (75)	3001 (41.3)	0.313
Abdominal pain or bloating	817 (11.2)	26 (68.4)	791 (10.9)	<0.001	2 (50)	815 (11.2)	0.065
Itching	925 (12.7)	16 (42.1)	909 (12.6)	<0.001	1 (25)	924 (12.7)	1
Rash, hives	1106 (15.2)	10 (26.3)	1096 (15.1)	0.068	1 (25)	1105 (15.2)	1
Skin’s sensitivity to the sun	274 (3.8)	3 (7.9)	271 (3.7)	0.171	0 (0)	274 (3.8)	1
Nausea, heartburn	430 (5.9)	15 (39.5)	415 (5.7)	<0.001	1 (25)	429 (5.9)	0.216
Cold hands and feet	808 (11.1)	17 (44.7)	791 (10.9)	<0.001	1 (25)	807 (11.1)	0.376
Swelling sensation in the hands	24 (0.3)	2 (5.3)	22 (0.3)	0.007	0 (0	24 (0.3)	1
Palpitations	139 (1.9)	8 (21.1)	131 (1.8)	<0.001	1 (25)	138 (1.9)	0.074
Chest pain	169 (2.3)	15 (39.5)	154 (2.1)	<0.001	2 (50)	167 (2.3)	0.003
Dizziness	120 (1.6)	8 (21.1)	112 (1.5)	<0.001	1 (25)	119 (1.6)	0.064
Shortness of breath	163 (2.2)	11 (28.9)	152 (2.1)	<0.001	1 (25)	162 (2.2)	0.087
Cramps	256 (3.5)	15 (39.5)	241 (3.3)	<0.001	1 (25)	255 (3.5)	0.134
Muscle weakness in the legs	311 (4.3)	23 (60.5)	288 (4)	<0.001	0 (0)	307 (4.2)	<0.001
Mild fever	891 (12.2)	14 (36.8)	877 (12.1)	<0.001	2 (50)	889 (12.2)	0.076
Painful menstruation	172 (2.3 *)	1 (31.6 ^#^)	162 (2.2 °)	<0.001	0 (0)	174 (2.4 ^)	1
Muscle stiffness, especially in the morning	31 (0.4)	5 (13.2)	26 (0.4)	<0.001	1 (25)	30 (0.4)	0.017
Shaking of the legs in bed	289 (4)	14 (36.8)	275 (3.8)	<0.001	2 (50)	287 (3.9)	0.009
Tingling	184 (2.5)	11 (28.9)	173 (2.4)	<0.001	0 (0)	184 (2.5)	1
Numbness	69 (0.9)	5 (13.2)	64 (0.9)	<0.001	0 (0)	69 (0.9)	1

Among females older than 10 years: * 17.3%; ^#^ 80%; ° 16.3%; ^ 17.3%.

**Table 4 biomedicines-11-01583-t004:** Association between the classification of the children according to the criteria for fibromyalgia (2010 ACR) and the contextual factors.

Variables		Total		Among Whom Matched the Criteria for Fibromyalgia			
		*n*	%	2010 ACR		2016 ACR	
				*n* (%); (Total: 38)	*p* *	*n* (%); (Total: 4)	*p* *
Relationship with peers	Very good	4380	60.2	9 (23.7%)	<0.001	1 (25%)	0.242
	Quite good	2708	37.2	19 (50%)		3 (75%)	
	Quite bad	140	1.9	9 (23.7%)		0	
	Very bad	8	0.1	0		0	
	Don’t know	39	0.5	1 (2.6%)		0	
School performance	Very good	3535	48.6	7 (18.4%)	<0.001	1 (25%)	0.558
	Quite good	3052	42	20 (52.6%)		3 (75%)	
	Quite bad	238	3.3	8 (21.1%)		0	
	Very bad	11	0.2	2 (5.3%)		0	
	Don’t know	439	6.0	1 (2.6%)		0	
Being bullied	Yes	821	11.3	17 (44.7%)	<0.001	1 (25%)	0.593
	No	5810	79.9	17 (44.7%)		3 (75%)	
	Don’t know	644	8.9	4 (0.6%)			
Cohabiting parents	No	1783	24.5	4 (10.5%)	0.05	0	0.325
	Yes	5492	75.5	34 (89.5%)			
Relationship with parents	Quite or very bad with at least one parent	93	1.3	8 (21.6%)	<0.001	0	0.950
At least one parent with diffuse pain	Yes	2023	27.8	32 (84.2%)	<0.001	4 (100%)	0.006
At least one parent with fibromyalgia	Yes	187	2.6	6 (15.8%)	<0.001	2 (50%)	0.004
At least one parent who uses or has used psychotropic drugs	Yes	659	9.1	10 (26.3%)	0.002	0	0.684
Road accident	Yes	205	2.8%	1 (2.6%)	0.709	0	0.892
Injury in the leisure time	Yes	493	6.8%	17 (44.7%)	<0.001	1 (25%)	0.245
Surgery	Yes	856	11.8%	12 (31.6%)	<0.001	1 (25%)	0.394

* Fisher exact test; ACR: American College of Rheumatology.

**Table 5 biomedicines-11-01583-t005:** Multivariate logistic regression analysis. Outcome (dependent) variable: agreement with 2010 ACR criteria (yes vs. no). OR: odds ratio.

Independent Variables		OR	Standard Error	*p*
Age	<8	1	-	-
	≥8	2.42	0.428	0.039
Injury in the leisure time	no	1	-	-
	yes	6.49	0.34	<0.001
At least one parent with fibromyalgia	no	1	-	-
	yes	2.54	0.469	0.046
At least one parent with diffuse pain	no	1	-	-
	yes	9.09	0.458	<0.001

**Table 6 biomedicines-11-01583-t006:** Distribution of somatic symptoms (%) in children with JFM in our study compared with that reported by other authors.

Symptoms	Our Data (2010 ACR Criteria)	Wolfe 1990 [4] (1990 ACR Criteria)	Yunus 1985 [6] (Yunus and Masi 1985 Criteria)	Siegel 1998 [22] (1990 ACR Criteria)	Eraso 2007 [20] Specifically Designed Questionnaire		Gedalia 2000 [23] (1990 ACR Criteria)
	*n* = 38	*n* = 45 Adults	*n* = 33	*n* = 33	*n* = 46 Onset 3–10 Years	*n* = 102 Onset 10–19 Years	*n* = 59
Diffuse pain	100	98	97	93	100	100	97
Fatigue	94.7	81	91	62	28	23	20
Sleep disturbances	68.4	75	67	96	65	74	70
Headache	78.9	53	58	71	78	80	76
Depression	57.9	32	55	43	9	9	7
Abdominal symptoms	26.2	30	27	38	39	19	17
Anxiety and nervousness	86.8	48	70	22	2	2	-
Subjective soft tissue swelling	5.3		61	40	39	14	24
Dysmenorrhea	31.6 *	41	-	36	-	-	-
Stiffness, especially in the morning	13.2	77	79	53	39	21	30
Numbness	13.2	67	36	24	6	1	-

* 80 among females older than 10 years.

## Data Availability

Data are available for scientific purposes upon request to the corresponding author.

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
