# Peer review of "Prevalence and Contextual Factors of Juvenile Fibromyalgia in a Population-Based Italian Sample of Children and Adolescents"

_biomedicines, 2023, doi:10.3390/biomedicines11061583_

Round 1

Reviewer 1 Report

I was pleased to read the manuscript entitled "Prevalence and contextual factors of Juvenile Fibromyalgia in a population-based Italian sample of children and adolescents" and to review it.

The study examined the prevalence of Juvenile Fibromyalgia (JFM) in a population-based sample of children and adolescents in Italy and assessed the relationships between JFM and contextual factors of the children and their parents. From a scientific point of view, it was interesting to read the article as it is one of the few studies that was focussed on a rare but very serious disease in childhood. The article is written in a typical format and is well structured. In my opinion, the content of the article is reasonable and comprehensive, but some corrections may be appropriate.

Keywords: The list of keywords should be substantially revised and such essential keywords as 'fibromyalgia' and 'children' should be included.

Line 164: Check if correct ‘SS>=9’.

Lines 188-199: The description of variables may be abbreviated, as the categories of significant variables are listed in Table 4.

Thank you for considering my opinion. I encourage authors to keep on working to improve the manuscript.

Moderate editing of English language.

Author Response

Dear Reviewer,

thank for the interest in our manuscript. We have taken into account you valuable suggestions. 

Kind regards

Chiara Lorini

Reviewer 2 Report

The paper presents an empirical study investigating the prevalence and contextual factors of Juvenile Fibromyalgia in a population-based Italian sample of children and adolescents.

The manuscript is well written and understandable.

The research topic is of relevance for both research and practice.

Important strengths include the detailed assessments and the relevance for health management in early life.

The paper could nicely fit into the Special Issue Advanced Research on Fibromyalgia.

However, the sample of around 50 individuals is relatively small. Has an a priori power analysis been conducted? Are estimates reliable with such a small sample?

The conceptual rationale could be elaborated in more depth. How are current models advanced?

The practical relevance could be illustrated with more detailed examples from everyday life. How will this study help to change the life and health of the targeted population in detail?

Author Response

(The authors gave the same response as above.)

Reviewer 3 Report

The study tells that JFM is a condition that is often overlooked in children, and early diagnosis is crucial. The study suggests that the 2010 ACR criteria are more appropriate for diagnosing JFM in children and adolescents. Additionally, the study emphasizes the need for family pediatricians to be aware of important predictors of JFM and to consider contextual factors when evaluating children with persistent pain.

I have some suggestions:

1. The introduction can be more concentrated. There are too many paragraphs.

2. The number of IRB approval should be given at the methodology session.

3. A table is suggested to present the Questionnaire.

4. Regarding the multivariate analysis, please specify which is the independent/dependent variable.

Author Response

(The authors gave the same response as above.)
